# Voltage-Gated Sodium Channel Na_V_1.7 Inhibitors with Potent Anticancer Activities in Medullary Thyroid Cancer Cells

**DOI:** 10.3390/cancers15102806

**Published:** 2023-05-17

**Authors:** Piyasuda Pukkanasut, Jason Whitt, Rachael Guenter, Shannon E. Lynch, Carlos Gallegos, Margarita Jacaranda Rosendo-Pineda, Juan Carlos Gomora, Herbert Chen, Diana Lin, Anna Sorace, Renata Jaskula-Sztul, Sadanandan E. Velu

**Affiliations:** 1Department of Chemistry, The University of Alabama at Birmingham, Birmingham, AL 35294, USA; piyasuda@uab.edu; 2Department of Surgery, The University of Alabama at Birmingham, Birmingham, AL 35294, USA; jwhitt@uabmc.edu (J.W.); rguenter@uab.edu (R.G.); hchen@uabmc.edu (H.C.); 3Graduate Biomedical Sciences, The University of Alabama at Birmingham, Birmingham, AL 35294, USA; slynch97@uab.edu; 4Department of Radiology, The University of Alabama at Birmingham, Birmingham, AL 35294, USA; carlosgt@uab.edu (C.G.);; 5Department of Biomedical Engineering, The University of Alabama at Birmingham, Birmingham, AL 35294, USA; 6Departamento de Neuropatología Molecular, Instituto de Fisiología Celular, Universidad Nacional Autónoma de México, Mexico City 04510, Mexico; mrosendo@ifc.unam.mx (M.J.R.-P.); jgomora@ifc.unam.mx (J.C.G.); 7Department of Pathology, The University of Alabama at Birmingham, Birmingham, AL 35233, USA; dmlin@uab.edu; 8O’Neal Comprehensive Cancer Center, The University of Alabama at Birmingham, Birmingham, AL 35233, USA

**Keywords:** voltage-gated sodium channels, Na_V_1.7, neuroendocrine tumor, medullary thyroid cancer, metastasis, cell invasion, cell viability

## Abstract

**Simple Summary:**

Despite the recent advances in the diagnosis and treatment of medullary thyroid cancer (MTC), it remains an understudied cancer type and continues to disproportionately contribute to thyroid-cancer-related mortality. In this manuscript, we report, for the first time, the overexpression of voltage-gated sodium channel subtype Na_V_1.7 in MTC cells and MTC patient samples, which is not expressed in normal thyroid cells and tissues. We establish the druggability of this channel by identifying a novel inhibitor (**SV188**) of this channel and investigate its mode of binding and ability to inhibit the *I*_Na_ current in Na_V_1.7. We also show that **SV188** significantly inhibited the migration and invasion of aggressive MTC cells at doses lower than its cytotoxic concentration. Overall, our data suggest that the unique overexpression of Na_V_1.7 in MTC can be exploited for the discovery of novel small-molecule drugs to treat MTC metastasis.

**Abstract:**

Our results from quantitative RT-PCR, Western blotting, immunohistochemistry, and the tissue microarray of medullary thyroid cancer (MTC) cell lines and patient specimens confirm that VGSC subtype Na_V_1.7 is uniquely expressed in aggressive MTC and not expressed in normal thyroid cells and tissues. We establish the druggability of Na_V_1.7 in MTC by identifying a novel inhibitor (**SV188**) and investigate its mode of binding and ability to inhibit *I*_Na_ current in Na_V_1.7. The whole-cell patch-clamp studies of the **SV188** in the Na_V_1.7 channels expressed in HEK-293 cells show that **SV188** inhibited the *I*_Na_ current in Na_V_1.7 with an IC_50_ value of 3.6 µM by a voltage- and use-dependent blockade mechanism, and the maximum inhibitory effect is observed when the channel is open. **SV188** inhibited the viability of MTC cell lines, MZ-CRC-1 and TT, with IC_50_ values of 8.47 μM and 9.32 μM, respectively, and significantly inhibited the invasion of MZ-CRC-1 cells by 35% and 52% at 3 μM and 6 μM, respectively. In contrast, **SV188** had no effect on the invasion of TT cells derived from primary tumor, which have lower basal expression of Na_V_1.7. In addition, **SV188** at 3 μM significantly inhibited the migration of MZ-CRC-1 and TT cells by 27% and 57%, respectively.

## 1. Introduction

Medullary thyroid cancer (MTC) is a type of neuroendocrine tumor (NET) evolving from neural-crest-derived calcitonin-producing parafollicular C cells, which in turn are responsible for controlling Ca^2+^ levels in the bloodstream [1,2,3]. MTC accounts for approximately 4% of all thyroid cancer cases but disproportionally accounts for 13% of thyroid-cancer-related deaths [4,5]. The major cause of MTC deaths is hepatic metastases, and patients with the metastatic form of this disease have a poor prognosis, with a 10-year survival rate of only 10% [6]. This subtype within thyroid cancer is particularly challenging to treat, as it does not respond to standard-of-care treatments [7]. Surgery is the only curative treatment for MTC [8]. Although, there are targeted agents to treat metastatic disease, none show an effect on overall survival [9]. Tyrosine kinase inhibitors (TKIs) are one of the treatment options for metastatic MTC [10]. Currently, there are only four FDA-approved TKIs that have been used for the treatment of advanced or progressive MTC, namely, vandetanib, cabozatinib, selpercatinib, and pralsetinib [4]. Although these are considered to be promising drugs to treat advanced MTC, drug resistance arising from the mutations in tyrosine kinase domains is a significant problem [11]. In addition, these drugs exhibit multiple side effects such as diarrhea, rash, fatigue, hypertension, and weight loss [12]. Furthermore, approximately 27% of vandetanib-treated patients showed QTc (corrected for heart rate) prolongation, which is a serious side effect that could lead to sudden cardiac arrest [12,13]. Despite the recent advances in diagnosis and treatment, MTC remains an understudied cancer type and continues to disproportionately contribute to thyroid-cancer-related mortality. Therefore, there is a need in the field to identify additional therapeutic targets for MTC.

One such promising anti-metastatic drug target is a family of voltage-gated sodium channels (VGSCs) [14,15,16,17], which are responsible for the generation and propagation of action potentials in excitable cells [18,19]. There are nine different subtypes of VGSCs expressed in different organs, namely, Na_V_1.1–Na_V_1.9. The complex structure of VGSCs consists of an α-subunit and one or two auxiliary β-subunits. The α-subunit contains four very similar domains, and each domain contains six transmembrane domains, S1–S6, where S1–S4 are the voltage-sensing domains and S5 and S6 are the pore-forming domains (Figure 1A,B) [20]. VGSCs play a crucial role in the membrane depolarization during the action potential in excitable cells such as neurons, skeletal, and cardiac muscle cells. The effect of the membrane potential (*V*_m_) in non-excitable cells such as cancer cells was first discovered in the 1970s. Although a few studies reported the excitability of cancer cells where a larger number of membrane currents and *V*_m_ fluctuations was observed [21,22], these discrete fluctuations might not be enough to generate or propagate the action potentials, a distinctive characteristic that defines a cell as “electrically excitable” [23]. The changes in the *V*_m_ in cancer cells and other non-excitable cells were found to be related to cell proliferation [24], migration [25], and wound healing and regeneration [26,27]. According to Tokuoka et al., membrane potential becomes significantly less negative during the transformation of normal cells to cancerous [28]. Similarly, several studies showed that the *V*_m_ in cancer cells is more depolarized and that cancer cells have substantially higher intracellular Na^+^ levels compared to non-cancerous tissues [28,29,30,31].

Recent studies have put forth a few plausible mechanisms for the involvement of VGSC sub-types in the development of metastasis in various tumors [14,32,33,34]. VGSCs are co-localized with the Na^+^/H^+^ exchanger isoform 1, the NHE1 and Na^+^/Ca^2+^ exchanger, and the NCX in the cell membrane [33,35,36]. An increase in Na^+^ influx activates H^+^ efflux through NHE1, thereby increasing the acidity of the tumor microenvironment. An acidic tumor microenvironment is known to activate the secretion of extracellular matrix proteases, most notably cathepsins and matrix metalloproteases (MMPs), which facilitate cancer cell migration from the primary tumor to the distal metastatic sites [37,38]. At the same time, an increased Na^+^ concentration within the cells results in a Ca^2+^ influx through NCX activation that leads to a higher Ca^2+^ consumption by the mitochondria, which then release Ca^2+^ to the cytosol. A greater Ca^2+^ concentration in the cytosol initiates actin polymerization and the formation of invadopodia, which supports cancer cell movement and migration (Figure 1C) [14,33,36]. Thus, VGSCs play a critical role in promoting tumor metastasis; therefore, the inhibition of VGSC activity by small molecules is a novel strategy for the development of therapeutic drugs for metastatic cancers [39,40,41].

**Figure 1 cancers-15-02806-f001:**
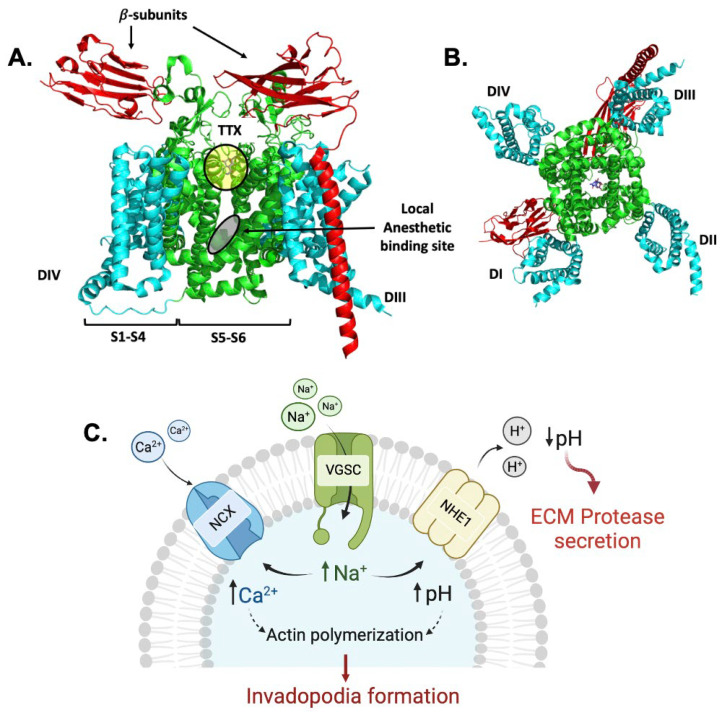
Schematic structure of the voltage-gated sodium channel subtype Na_V_1.7 (PDB 6j8j) [42] with 𝛽-subunits and tetrodotoxin (TTX). (**A**) The side view of Na_V_1.7 transmembrane segments S1–S6; S1–S4 are shown in cyan, S5–S6 are shown in green, TTX binding site is highlighted by yellow circle, and local anesthetic binding site is highlighted by gray oval. (**B**) The bottom view of Na_V_1.7 with TTX bound. (**C**) The proposed mechanism for the involvement of VGSCs in cancer cell motility; VGSC is colocalized with NHE1 and NCX. The activity of VGSC facilitates cancer cell motility by increasing acidity of extracellular matrix (ECM) environment, inducing ECM protease secretion, and increasing the concentration of Ca^2+^ in intracellular fluid, which supports invadopodia formation by cancer cells.

VGSCs are druggable targets, and their inhibitors are commonly used as anticonvulsants, local anesthetics, and antiarrhythmics and in the treatment of neuronal excitability disorders [43]. Clinically used VGSC inhibitors are considered to be state-dependent or use-dependent inhibitors, which show higher affinity to the binding site when the channel is in the open or inactivated state and show lower affinity when the channel is in the resting state [44,45]. The selectivity of drugs toward the cells in the disease state vs. the normal state is due to the preferential binding of the drug molecules to the binding site (the S6 of Domain IV), which is located in the inner pore of the channel (Figure 1A). In the disease state, the channels have higher rates of cell depolarization (opening state); as a consequence, the relative time of the channels staying in the resting state is lower than that of the normal cells, resulting in more selective drug binding to the channels in the disease state [46,47,48].

In recent years, VGSC expression has been found to be aberrantly enhanced in non-excitable cells in aggressive human cancers of epithelial origin such as lung, prostate, ovarian, colon, and breast cancer, and this overexpression has been shown to be associated with cancer cells invasiveness [14,16,26,41,49,50]. To date, multiple VGSC subtypes have been targeted for the discovery of potential anticancer drugs [15,16,17,40,51,52]. Recently, Na_V_1.6 was found to promote human follicular thyroid carcinoma by increasing cell proliferation, epithelial-to-mesenchymal transition, and invasion [53]. However, no such investigation of the sodium channel inhibitors in MTC has been reported since the initial discovery of the presence of sodium channel genes in MTC cells by Klugbauer et al. in 1995 [54].

As a part of our interest in targeting VGSCs for cancer therapy, we recently reported small-molecule inhibitors for the VGSC subtype Na_V_1.5, with impressive cell invasion-inhibitory activities in breast cancer cells, MDA-MB-231 and colon cancer cells, SW620 and HCT116 [40,55]. As a continuation of these studies, we investigated the expression of VGSC subtype Na_V_1.7 in MTC and discovered the small-molecule inhibitors of this channel. Here, we report, for the first time, the discovery of the overexpression of Na_V_1.7 (*SCN9A* gene) in aggressive MTC cells and patient samples and the lack of this protein in normal thyroid cells and tissues. We further establish the druggability of the Na_V_1.7 channel in MTC by identifying a novel inhibitor and investigate its mode of binding and ability to inhibit the Na^+^ current (*I*_Na_) in Na_V_1.7. This study demonstrates how the lead compound targeting Na_V_1.7 inhibits MTC cell viability, migration, and invasion in vitro.

## 2. Materials and Methods

### 2.1. Cell Culture

Human MTC cell line MZ-CRC-1 was obtained from Dr. Gilbert Cote (MD Anderson Cancer Center, Houston, TX, USA) and human MTC cell line (TT) was obtained from Barry D. Nelkin (John Hopkins University, Baltimore, MD, USA). Human MTC cell lines were maintained under the condition described in [56]. Mouse MTC cell line (MTC-p25OE) was obtained from James Bibb (University of Alabama at Birmingham, Birmingham, AL, USA). Human normal thyroid cell lines (Htori-3 and Nty-ori) were purchased from Sigma Life Science/European Collection of Cell Cultures.

### 2.2. Human Tissue Samples

Human MTC tumor samples with pathology status and control tumor samples were obtained from UAB Tissue Biorepository, with approved IRB protocol (IRB-300006132-002). The MTC microarray contained formalin-fixed, paraffin-embedded thyroid biopsies from 45 patients including normal thyroid, primary MTC, and metastatic MTC, each mounted in triplicate for a total of 133 cores. Tumor cell lysates were prepared for Western blot analysis as described below.

### 2.3. Western Blot Analysis

Cells or tumor specimens were lysed using radio-immunoprecipitation assay (RIPA) buffer with the addition of protease and phosphatase inhibitor (Sigma-Aldrich, St. Louis, MO, USA). Protein concentrations in each sample were quantified using a Pierce BCA Protein Assay Kit (Thermos Scientific, Waltham, MA, USA). Prior to performing gel electrophoresis, 1:1 of 2× Leammli Sample buffer (Bio-Rad, Hercules, CA, USA) was added to protein samples. The mixture was diluted with 5% 2-mercaptoethanol (ThermoFisher Scientific, Waltham, MA, USA). All protein samples were heated at 95 °C for 5 min and run on 4–15% Criterion TGX gradient gels (Bio-Rad). Gel transfer and immunoblotting detections were performed as previously described [57]. Primary antibodies for Na_V_1.7 (EMD Millipore Corp., Darmstadt, Germany) were used at 1:1000, and the reference protein GAPDH (Cell Signaling Technology, Danvers, CA, USA) and β-actin (Cell Signaling Technology) were used at 1:2000. Horseradish peroxidase-conjugated anti-rabbit/mouse with a dilution of 1:1000 (Cell Signaling Technology) was used as secondary antibodies. The molecular weight-marker broad-range protein ladder (10–260 kDa) (Spectra Multicolor, ThermoFisher Scientific) was used to confirm the size of the protein of our interest.

### 2.4. TMA Staining, Quantification, and Evaluation

Na_V_1.7 was immunostained using an anti-Na_V_1.7 monoclonal antibody (ab85015, Abcam, Cambridge, UK). Positive and negative immunohistochemistry (IHC) controls were generated from cell lines that represent high or no expression of Na_V_1.7. MZ-CRC-1, which showed high expression of Na_V_1.7, was used as a positive control, and Nthy-ori3-1 (normal thyroid), which did not have an expression of Na_V_1.7, was used as a negative control. Na_V_1.7 expression was quantified within each core using an automated digital quantification (custom MATLAB code). IHC samples were automatically segmented to extract tissue boundaries and transition from RGB images to HSV, followed by a saturation mask to distinguish tissue. The distribution of saturation was plotted, and Otsu’s automated threshold for separating positive vs. negative staining was employed to extract out percentage of positive tissue expression.

### 2.5. Real-Time Quantitative PCR (RT-qPCR)

Each RNA sample was isolated using RNeasy Plus Mini kit (Qiagen, Hilden, Germany). The RNA concentrations were determined by NanoDrop 1000 spectrophotometer (Thermo Fisher Scientific). RNA samples that have a ratio of absorbance greater than 2.0 at 260 nm and 280 nm were used in the experiments. Complementary DNA (cDNA) was synthesized using iScript RT Supermix (Bio-Rad), and 1 μg total RNA was used in each sample. PCR samples were prepared using SYBR Green master mixes (Bio-Rad). Real-time quantitative PCR was performed in triplicate on CFX Connect Real-Time PCR Detection System (Bio-Rad). The sequences of the PCR primers, *SCN5A* (Na_V_1.5), *SCN8A* (Na_V_1.6), *SCN9A* (Na_V_1.7), and *SCN9A1* (NHE1) used for the analysis in this experiment are *SCN5A* (Na_V_1.5) forward: CACGCGTTCACTTTCCTTC, reverse: CATCAGCCAGCTTCTTCACA, *SCN8A* (Na_V_1.6) forward: CGCCTTATGACCCAGGACTA, reverse: GTGCCTCTTCCTGTTGCTTC, *SCN9A* (Na_V_1.7) forward: GGCTCCTTGTTTTCTGCAAG, reverse: TGGCTTGGCTGATGTTACTG and *SCN9A1* (NHE1) forward: GGCATCGAGGACATCTGTGG, reverse: CTGCAGACTTGGGGTGGATG, as described in [34]. Target gene expression was normalized to either S27 or GAPDH, and the ΔΔCt method was used to calculate relative gene expression [58]. Error bars show the standard error of the mean (SEM).

### 2.6. Na_V_1.7 Transfection

Human embryonic kidney cells (HEK-293) were acquired from the American Type Cell Culture Collection (ATCC CRL-1573) and grown in DMEM/F12 mixture supplemented with 10% FBS, 100 U/mL penicillin, and 100 µg/mL streptomycin at 37 °C in a CO_2_ incubator. Transient transfections were performed with PEI (polyethylenimine; Santa Cruz Biotechnology, Dallas, TX, USA) in 35 mm dishes, by using a 3:1 ratio for PEI:DNA. HEK-293 cells were transfected with 2.5 µg of rat cDNA Na_V_1.7 (GenBank No. U79568) and 0.2 µg of GFP cDNA as a reporter gene. After transfection, cells were cultured for 24–72 h before being dissociated and seeded on 0.25 cm^2^ glass coverslips contained in a 35 mm Petri dish for electrophysiological experiments.

### 2.7. Electrophysiology

Sodium currents (*I*_Na_) of Na_V_1.7 channels were recorded at room temperature (21 ± 2 °C) with the whole-cell configuration of the patch-clamp technique [59,60]. The Na_V_1.7 channels’ activity was investigated by using an Axopatch 200B amplifier, a Digidata1550B A/D converter, and pCLAMP 10.7 software (Molecular Devices, San Jose, CA, USA). Unless otherwise noted, the holding potential (HP) used in the experiments was −120 mV. Current recordings were usually sampled at 50 kHz, following 5 kHz analogue filtering. Whole-cell series resistance (*R*_s_) and cell capacitance (*C*_m_) were estimated from optimal cancellation of the capacitive transients with the built-in circuitry of the amplifier, and in some cases *R*_s_ was compensated electrically by 60–80%. Currents were recorded on two channels: one with on-line leak subtraction using the P/-5 method, and the other to evaluate cell stability and holding current. Only leak-subtracted data are shown. Recording pipettes were made from TW150-3 capillary tubing (WPI, Inc., Sarasota, FL, USA), using a Model P-97 Flaming-Brown pipette puller (Sutter Instrument Co., Novato, CA, USA). Cells were bathed in a solution containing the following composition (in mM): 158 NaCl, 2 CaCl_2_, 2 MgCl_2_, and 10 HEPES-NaOH (pH 7.4), with an osmolality of 305–310 mOsm. Cells were patched with microelectrodes containing the following internal solution (in mM): 110 CsF, 30 NaCl, 2 CaCl_2_, 10 EGTA, and 10 HEPES-CsOH (pH 7.4), with an osmolality of 295–300 mOsm. The recording chamber was continuously perfused by gravity at a rate of 2 mL/min, and solution exchange was accomplished by a manually controlled six-way rotary valve. A 50 mM stock solution of the compound **SV188** dissolved in DMSO was used to prepare fresh test concentrations in external solution ranging from 0.3 to 30 µM. The highest concentration of DMSO in the tested **SV188** solutions was 0.06%. Voltage-gated sodium currents were monitored by 16-ms depolarizing pulses to −10 mV from an HP of −120 mV applied every 10 s. Modifications to this protocol were used to obtain data concerning current–voltage (*I–V*) relationships and steady-state inactivation of sodium channels. Peak current values of current recordings were obtained by using the Clampfit application of pCLAMP software. Dose–response relationships for **SV188** blockade were fit with the following Hill equation: Y = 1/(1 + 10 [(log IC_50_ − X) × *h*]), where X is the logarithm of concentration, Y is the fraction of current remaining after addition of the drug, IC_50_ is the concentration required for 50% blockade of current, and *h* is the Hill coefficient. For this analysis, current in control external solution was normalized to 1, and we assumed complete blockade of current with sufficient drug concentration. The voltage dependence of current activation was described with a single Boltzmann distribution: *G* = *G*_max_/(1 + exp (−(*V*_m_ − *V*_1/2_)/*k*)), where *G*_max_ is the maximum normalized Na^+^ conductance, *V*_m_ is the test potential, *V*_1/2_ is the midpoint of activation, and *k* is the slope factor. The voltage-dependence of steady-state inactivation was also described with a single Boltzmann function as follows: *I* = *I*_max_/(1 + exp ((*V*_m_ − *V*_1/2_)/*k*)), where *I*_max_ is the maximal normalized sodium current, *V*_m_ is the test potential, *V*_1/2_ is the midpoint of steady-state inactivation, and *k* is the slope factor.

### 2.8. Cell Viability Assay

A 3-(4,5-Dimethylthiazole-2-yl)-2,5-diphenyl tetrazolium bromide (MTT) assay (Sigma-Aldrich) was used to measure the effect of the compound against cell proliferation and IC_50_ determination. MTC cell lines were plated in flat-bottom 96-well plate at seeding density of 10^4^ cells/well. Cells were allowed to grow overnight. Different concentrations of the treatment up to 100 μM were tested compared to control (0.2% DMSO) and incubated for 72 h. After the incubation, cell viability was determined using MTT reagent, the treatment media was removed, and 25 µL of serum-free media containing 0.5 mg/mL MTT (Sigma-Aldrich) was added to each well, followed by an incubation at 37 °C for 2 h. Then, the cells and MTT reagent were suspended with 75 µL of DMSO prior to the analysis at 562 nm using a plate reader (Infinite M200 PRO, TECAN, Männedorf, Switzerland). Percentage of cell viability and concentrations were plotted, and the IC_50_ value was calculated using GraphPad Prism9.3.1 [61]. The concentrations were converted to log_10_ (concentration), and the IC_50_ curve was plotted with normalized curve fit vs. dose response (variable slope) to obtain the IC_50_ value.

### 2.9. Motility Assays (Migration/Invasion)

Inhibition of cell migration and invasion was determined using the Boyden chamber assay. For migration assay, transwell cell culture inserts, with 8.0 μM pore size (Corning Life Sciences, Corning, NY, USA), were plated in 24-well plate (Costar, Corning Life Sciences). MTC cell suspension in serum-free media containing 0.06% DMSO (as a control) and different concentrations of **SV188** were plated (4 × 10^5^ cells per insert) in upper compartment of transwell cell culture inserts, with 8.0 μM pore size, while 650 µL of media containing fetal bovine serum (chemo-attractant) was added in 24-well plates. MTC cells were allowed to migrate for 48 h. The cells that migrated through the membrane were stained using three-step staining kit (Fisher, Rockingham County, NH, USA). The membrane was cut and mounted on microscope slides, with size 25 × 75 × 1 mm (Fisher). The membranes were covered using microscope cover glass (Fisher). Migrated cells were counted from microscope (OLYMPUS DP74) imaging. Number of migrated cells from each individual experiment was normalized, and triplicated results were reported as mean fold change in number of cells migrated through membrane ± SEM. For invasion assay, the upper compartment was coated with Matrigel Metrix (Corning Life Sciences) to mimic the extra cellular matrix (ECM) environment. The gel was allowed to set by incubation at 37 °C for 2 h. Number of invaded cells from each individual experiment was normalized and reported as mean fold change in number of cells migrated through membrane ± SEM. Migration/invasion in each concentration was completed in quadruplicate for a total of 3 experiments. The results were plotted, and statistical significance was determined using GraphPad Prism 9.3.1 [61].

### 2.10. Cell Cycle Analysis Flow Cytometry

Cell cycle analysis data were acquired using Flow Cytometry (BD LSRFortessa™, BD Biosciences, Franklin Lakes, NJ, USA), and at least 3000 events were collected in each sample for 3 individual experiments. MZ-CRC-1 cells were plated on 90 mm Petri dishes in suspension (0.5–1 × 10^6^ cells). Cells were allowed to grow overnight before changing to the treatment media containing different concentrations of **SV188**; the final concentration were 3 µM, 6 µM, and 9 µM, and growth media containing 0.09% DMSO was used for the control. Cells were incubated for 48 h before being harvested using buffer containing EDTA. The cell pellets were washed with PBS prior to fixing by adding 70% ethanol (ice cold) dropwise while gently vortexing. The cells were fixed at −20 °C overnight. The next day, ethanol was removed, and cells were washed again with PBS and resuspended in staining buffer containing propidium iodide (PI) and *RNase* (*FxCycle*^TM^PI/RNase Staining, Invitrogen, Waltham, MA, USA). Cells were incubated in staining buffer in a dark cold place (4 °C) for 30 min before transferring to a cell cycle analysis tube and acquiring the data using flow cytometry. The data were processed using FlowJo 10.8.1 [62]. The combined results were plotted, and statistical significance was determined using GraphPad Prism 9.3.1 [61].

### 2.11. Statistical Analysis

Bivariate correlation with confidence interval was achieved through IBM SPSS Statistics for Macintosh, Version 29.0.0 [63]. Statistical significance was assessed using GraphPad Prism 9.3.1 [61], One-way ANOVA followed by Dunnett’s multiple comparisons test (GraphPad Software, San Diego, CA, USA). All data are expressed as mean ± standard error of the mean (SEM), unless otherwise noted.

### 2.12. General Methods for Compound Synthesis and Characterization

Anhydrous solvents used for reactions were purchased in Sure-Seal™ bottles from Aldrich chemical company. THF and ether were freshly distilled over sodium/benzophenone. Other reagents were purchased from Sigma-Aldrich, Alfa Aesar, or Acros. Solvent evaporations were carried out under vacuum using a rotary evaporator (BUCHI). Thin-layer chromatography (TLC) was performed on aluminum-backed Si gel plates, with fluorescent indicator (20 × 20 cm F-254, 200 μm, Dynamic Adsorbents, Norcross, GA, USA). TLC spots were visualized by UV light at 254 and 365 nm or by using staining agents such as ninhydrin or KMnO_4_. Purification by column and flash chromatography was carried out using Si gel (32–63 μm, Dynamic Adsorbents), using the solvent systems as indicated. The NMR spectra were recorded on a Bruker DPX 400 spectrometer. The peak calibration was accomplished using TMS or the NMR solvent peaks as internal standard. The chemical shift (δ) values and coupling constants (*J*) were given in parts per million and in Hz, respectively. Mass spectra were recorded on an Applied Biosystems 4000 Q Trap instrument at the Mass Spectrometry Facility in the department of Chemistry and Biochemistry, University of Alabama, Tuscaloosa, AL. All compounds are >97% pure by HPLC (Appendix A). HPLC traces were performed on Shimadzu HPLC with the following parts/software: DGU-20A_3_ Prominence Degasser, FCV-11AL Valve Unit, 2× LC-20AD Prominence Liquid Chromatographs, SIL-20AC HT Prominence Auto Sampler, CBM-20A Prominence Communications Bus Module, SPD-M20A Prominence Diode Array Detector, CTO-20AC Prominence Column Oven, and LCsolution Version 1.22 SP1. Mobile Phase Buffer (60% MeCN/40% H_2_O/0.1% formic acid) was freshly prepared using HPLC grade reagents/solvents in a 500 mL volumetric flask and thoroughly degassed using the DGU-20A_3_ Prominence Degasser. Raw data from the HPLC chromatograms were exported as text files and plotted using GraphPad Prism 9.3.1.

#### 2.12.1. 4,4-Diphenylbutyric Acid (**2**)

To a solution of phenyl butyrolactone, **1** (0.5 g, 3.08 mmol) in anhydrous benzene (20 mL), anhydrous AlCl_3_ (0.62 g, 4.63 mmol) was added slowly, and the reaction mixture was stirred overnight under N_2_ atmosphere. When the reaction was complete as indicated by TLC (50% EtOAc in hexanes, Rf = 0.4), pH of the reaction mixture was adjusted to 1 using 1N. HCl. The reaction mixture was further diluted with distilled water (20 mL) and extracted with ether (3 × 20 mL). The combined organic layer was washed with water (2 × 30 mL) and brine (1 × 30 mL) and dried over Na_2_SO_4_. The drying agent was filtered off, and the filtrate was concentrated in a expression and patient disease pure 4,4-diphenylbutyric acid, **2** as a white solid (0. 698g, 94%). mp: 102–103 °C; ^1^H-NMR (CDCl_3_, 400MHz) δ 2.19–2.23 (m, 2H), 2.26–2.32 (m, 2H), 3.84 (t, 2H, *J* = 7.4 Hz), 7.06–7.10 (m, 2H), 7.12–7.20 (m, 8H), 9.47 (brS, 1H); ^13^C-NMR (CDCl_3_) δ: 30.4, 30.7, 50.5, 126.6, 128.0, 128.7, 144.1, 179.9; HRMS [M-H]^+^ calculated for C_16_H_15_O_2_ 239.1072, found 239.1074.

#### 2.12.2. 3-(Piperidin-1-yl)propan-1-amine (**5**)

To a solution of 1-piperidinepropionitrile, **3** (1 g, 7.23 mmol) in anhydrous MeOH (100 mL), Raney-Ni (2.5 g) suspension in water was added quickly and stirred for 12 h at room temperature under a hydrogen atmosphere from a balloon. TLC examination (20% MeOH in CHCl_3_, Rf = 0.12) showed that the reaction was complete. Raney Ni was filtered off carefully over celite 545 and washed with MeOH (50 mL) continuously without letting the celite dry out. The combined filtrate was concentrated under vacuum, redissolved in CH_2_Cl_2_ (50 mL), and dried over Na_2_SO_4_. The drying agent was removed by filtration, and the filtrate was concentrated under vacuum to obtain 3-(piperidin-1-yl)propan-1-amine, **5** (0.914 g, 89%) as a colorless oil. ^1^H-NMR (CDCl_3_, 400 MHz) δ 1.52–1.58 (m, 6H), 1.63 (quint, 2H, *J* = 7.44 Hz), 2.33 (t, 6H, *J* = 7.24 Hz), 2.65 (brS, 2H), 2.73 (t, 2H, *J* = 6.76 Hz); ^13^C-NMR (CDCl_3_, 400 MHz) δ 24.2, 25.8, 27.5, 40.9, 54.4, 57.6, 76.8, 77.2, 77.5; HRMS [M + H]^+^ calculated for C_8_ H_19_ N_2_ 143.1548, found 143.1543.

#### 2.12.3. 4,4-Diphenyl-N-[3-(piperidin-1-yl)propyl]butanamide (**6**)

To a solution of 3-(piperidin-1-yl)propan-1-amine. **5** (0.88g, 6.20 mmol) in 200 mL CH_2_Cl_2_ (200 mL) 4,4-diphenylbutyric acid, **2** (1.79 g, 7.44 mmol) was added, followed by the addition of EDC (1.44 g, 9.30 mmol) and DMAP (0.76g, 0.62 mmol). The reaction mixture was stirred overnight at room temperature under N_2_ atmosphere. The TLC examination (10% MeOH in CHCl_3_, Rf = 0.3) indicated the completion of the reaction. The reaction mixture was washed with saturated NaHCO_3_ (2 × 50 mL), water (2 × 50 mL), and brine (1 × 50 mL) and dried over Na_2_SO_4_. The drying agent was removed by filtration, and the filtrate was concentrated under vacuum to obtain the crude product, which was purified by column chromatography over Si gel using 0–5% MeOH in CHCl_3_ as eluent to afford the pure product 4,4-Diphenyl-N-[3-(piperidin-1-yl)propyl]butanamide, **6** (1.88g, 83.2%) as a white solid. mp: 74 °C; ^1^H-NMR (CDCl_3_, 400 MHz) δ 1.41–1.47 (m, 6H), 1.63 (quint, 2H, *J* = 6.12 Hz), 2.10 (t, 2H, *J* = 8.64 Hz), 2.37–2.43 (m, 8 H), 3.30 (q, *J* = 5.36 Hz), 3.93 (t, 1H, *J* = 7.92 Hz), 7.15–7.19 (m, 2H), 7.24–7.30 (m, 8H), 7.45 (brS, 1H); ^13^C-NMR (CDCl_3_, 400 MHz) δ 24.3, 24.7, 26.1, 31.4, 35.4, 40.0, 50.7, 54.5, 58.6, 126.3, 127.9, 128.5, 144.4, 172.4; HRMS [M-H]^+^calculated for C_24_H_33_N_2_O 365.2593, found 365.2585.

#### 2.12.4. 4,4-Diphenylbutyl [3-(piperidin-1-yl)propyl]amine Hydrochloride (**SV188**)

To a solution of 4,4-diphenyl-N-[3-(piperidin-1-yl)propyl]butanamide, **6** (2.09 g, 5.73 mmol) in anhydrous THF (100 mL), LiAlH_4_ (0.87 g, 22.93 mmol) was added slowly under N_2_ atmosphere. The reaction mixture was refluxed for 2 h. TLC examination (10% MeOH in CHCl_3_) indicated the completion of the reaction. The reaction mixture was then carefully quenched by a very slow drop-wise addition of saturated Na_2_SO_4_ solution until the evolution of H_2_ ceased. The reaction mixture was then filtered over celite 545 and washed with EtOAc (100 mL). The combined filtrate was concentrated under vacuum, redissolved in EtOAc (100 mL), and dried over Na_2_SO_4_. The drying agent was filtered off, and the filtrate was concentrated under vacuum to obtain the amine product as a light-yellow oil. This product was dissolved in ether (10 mL) and treated with 2N. HCl (0.4 mL) to make the hydrochloride salt of 4,4-Diphenylbutyl [3-(piperidin-1-yl)propyl]amine hydrochloride, **SV188** as a white solid (1.47 g, 61%). mp: 215 °C (decomposed); ^1^H-NMR (DMSO-d_6_, 400 MHz) δ 1.32–1.41 (m, 1H), 1.54 (quint, 2H, *J* = 6.96 Hz), 1.66–1.84 (m, 5H), 2.06–2.10 (m, 4H), 2.77–2.92 (m, 6H), 3.09 (quint, 2H, *J* = 5.16 Hz), 3.33–3.40 (m, 2H), 3.94 (t, 1H, *J* = 7.88 Hz), 7.16 (t, 2H, *J* = 7.08 Hz), 7.26–7.33(m, 8H), 9.13 (brS, 2H), 10.72 (brS, 1H); ^13^C-NMR (DMSO-d_6_, 400 MHz) δ 19.8, 21.2, 22.0, 24.2, 31.6, 43.9, 46.5, 50.0, 51.7, 52.6, 126.0, 127.5, 128.3, 144.6; HRMS [M + H]^+^ calculated for C_24_H_35_N_2_ 351.2800, found 351.2792.

#### 2.12.5. 4,4-Diphenyl-N-(3-phenylpropyl)butanamide (**8**)

To a solution of 3-phenylpropylamine, **7** (0.5 g, 3.70 mmol) in CH_2_Cl_2_ (120 mL) 4-(4-phenyl)butyric acid, **2**, (0. 88 g, 3.70 mmol), EDC (0. 861 g, 5.55 mmol) and DMAP (0.045 g, 0.37 mmol) were added, and the reaction mixture was stirred at room temperature under N_2_ atmosphere overnight. The TLC examination (5% MeOH/in NH_3_-saturated CHCl_3_, Rf = 0.71) indicated the completion of the reaction. The reaction mixture was washed with saturated NaHCO_3_ (2 × 50 mL), water (2 × 50 mL), and brine (1 × 50 mL) and dried over Na_2_SO_4_. The drying agent was removed by filtration, and the filtrate was concentrated under vacuum to obtain the crude product, which was purified by column chromatography over Si gel using NH_3_-saturated CHCl_3_ as eluent to afford the pure 4,4-Diphenyl-N-(3-phenylpropyl)butanamide, **8** (0.967 g, 73%) as a yellow oil. ^1^H-NMR (CDCl_3_) δ 1.78 (quint, 2H, *J* = 7.32 Hz), 2.04 (t, 2H, *J* = 7.92 Hz), 2.33–2.38 (m, 2H), 2.61 (t, 2H, *J* = 7.76 Hz), 3.23 (q, 2H, *J* = 6.76 Hz), 3.89 (t, 1H, *J* = 7.96 Hz), 5.32 (brS, 1H), 7.16 (t, 5H, *J* = 7.40 Hz), 7.21–7.28 (m, 10H); ^13^C-NMR (CDCl_3_, 400 MHz) δ 31.1, 31.2, 33.3, 34.9, 39.1, 50.5, 126.0, 126.3, 127.8, 128. 3, 128.4, 128.5, 141.4, 144.2, 172.4; HRMS [M + H]^+^ calculated for C_25_H_28_NO 358.2171, found 358.2178.

#### 2.12.6. 4,4-Diphenylbutyl(3-phenylpropyl)amine Hydrochloride (**WJB-133**)

To a solution of 4,4-Diphenyl-N-(3-phenylpropyl)butanamide, **8** (0.5 g, 1.40 mmol) in anhydrous THF (35 mL), LiAlH_4_ (0.159 g, 4.2 mmol) was added slowly under N_2_ atmosphere. The reaction mixture was refluxed for 2 h. TLC examination (2% MeOH in NH_3_-saturated CHCl_3_) indicated the completion of the reaction. The reaction mixture was then carefully quenched by a very slow drop-wise addition of saturated Na_2_SO_4_ until the evolution of H_2_ ceased. The reaction mixture was then filtered over celite 545, and the filtrate was washed with EtOAc (100 mL). The combined filtrate was concentrated under vacuum, redissolved in EtOAc (100 mL), and dried over Na_2_SO_4_. The drying agent was filtered off, and the filtrate was concentrated under vacuum to obtain the amine product as a light-yellow oil. This product was dissolved in ether (4 mL) and treated with 2N. HCl (0.2 mL) to make the hydrochloride salt of **WJB-133** as a clear, gummy, sticky oil (0.248 g, 47%). ^1^H-NMR (CDCl_3_, 400 MHz) δ 1.65–1.80 (m, 2H), 2.03–2.09 (m, 4H), 2.56 (t, 2H, *J* = 6.60 Hz), 2.71–2.76 (m, 4H), 3.81 (t, 1H, *J* = 7.40 Hz), 7.10–7.19 (m, 10H), 7.21–7.26 (m, 5H), 9.47 (brS, 2H); ^13^C-NMR (CDCl_3_, 400 MHz) δ 24.0, 27.0, 32.5, 32.6, 46.4, 47.1, 50.7, 126.3, 126.4, 127.7, 128.3, 128.6 (2C), 139.7, 144.1; HRMS [M + H]^+^ calculated for C_25_H_30_N 344.2378, found 344.2380.

#### 2.12.7. 4-(4-Fluorophenyl)butyl][3-(piperidin-1-yl)propyl Amine Hydrochloride (Compound **4**)

4-(4-Fluorophenyl)butyl][3-(piperidin-1-yl)propyl]amine (compound **4**) was prepared following our previously reported procedure [40]. Compound **4** (0.042 g, 0.143 mmol) was converted to hydrochloride salt by the treatment of its solution in ether (2 mL) with 2N. HCl (0.1 mL) to obtain the hydrochloride salt of compound **4** (0.034 g, 72.4%) yield as a white solid. mp: 197 °C (decomposed); ^1^H-NMR (DMSO-d_6_, 400 MHz) δ 1.36–1.39 (m, 1H), 1.61–1.66 (m, 5H), 1.74–1.85 (m, 4H), 2.13 (t, 2H, *J* = 7.1 Hz), 2.58 (t, 2H, *J* = 7.2 Hz), 2.80–2.87 (m, 4H), 2.96 (t, 2H, *J* = 5.5 Hz), 3.10–3.15 (m, 2H), 3.35–3.38 (m, 2H), 7.07–7.11 (m, 2H), 7.23–7.27 (m, 2H), 9.28 (brS, 2H), 10.74 (brS, 1H); ^13^C-NMR (DMSO-d_6_, 400 MHz) δ 19.9, 21.4, 22.1, 24.9, 28.0, 33.6, 44.0, 46.5, 51.9, 52.8, F-splitting 114.8, 115.0, F-splitting 130.0, 130.1, F-splitting 137.7, 137.8, F-splitting 159.4, 161.8; HRMS [M + H]^+^ calculated for C_18_H_30_N_2_F 293.2393, found 293.2386.

## 3. Results and Discussion

### 3.1. VGSC Expression in Neuroendocrine Tumors (NETs)

The expression of VGSCs is reported to be associated with invasion and metastatic behavior of various cancers. A few examples of such channels are Na_V_1.5 in breast [40,51,52], colon [49], and ovarian cancers [64]; Na_V_1.6 in cervical cancer [65]; and Na_V_1.7 in prostate [66,67], gastric [34], lung [68], and endometrial cancers [69]. Over the past decade, VGSCs subtypes Na_V_1.5, Na_V_1.6, and Na_V_1.7 have been the most-reported isoforms that are shown to influence migration and invasion [32,41,70]. We initially examined the mRNA expression levels of VGSCs isoforms Na_V_1.5, Na_V_1.6, and Na_V_1.7 in NETs using pancreatic (BON), lung (H727), and thyroid (MZ-CRC-1 and TT) cells and observed that the aggressive MTC cells originated from lymph node metastasis; MZ-CRC-1 showed strong expression of channels Na_V_1.5, Na_V_1.6, and Na_V_1.7. The highest expression was observed in Na_V_1.7, which was 400-fold higher than the lowest expression of Na_V_1.5. Moreover, Na_V_1.7 was uniquely overexpressed in MTC cells, MZ-CRC-1, and TT compared to other NET cell lines, where MZ-CRC-1 was 1800-fold higher than BON and 30-fold higher than H727; TT was 700-fold higher than BON and 13-fold higher than H727. The highly metastatic MZ-CRC-1 cells showed two-fold higher expression of Na_V_1.7 compared to the weakly metastatic TT cells (Figure 2A), suggesting that the expression level of Na_V_1.7 could be correlated to the metastatic and aggressive behavior of MTC cell lines. Further, there was detectable expression of Na_V_1.5 among the less-aggressive NETs: MTC (TT), pancreatic cancer (BON), and lung cancer (H727) cells (Figure 2A). To further confirm this observation, we examined the mRNA expression of Na_V_1.7 in non-neuroendocrine thyroid cancers cells, normal thyroid cells, and MTC patient samples. We found that the expression of Na_V_1.7 is conserved in MZ-CRC-1 cells and in MTC patient tissues when compared to normal thyroid cells (Nthy-ori3-1 and Htori-3), normal thyroid counterparts (TH64 normal, TH79 normal, and TH46 normal), and cells that represent both papillary and anaplastic thyroid carcinomas (Figure 2B,C).

The expression of Na_V_1.7 in MTC cells and patient samples was also confirmed by immunoblotting. To establish the basal expression of Na_V_1.7 in MTC, we used MZ-CRC-1 and TT human cells; p25OE MTC cells originating from transgenic mice; and MTC patient tissues, cancerous and adjacent non-cancerous thyroid tissues, for direct comparison. We determined that only the MTC specimens, human MTC cell lines, and mouse transgenic MTC cells were Nav1.7 positive, (Figure 3A). The highly metastatic MZ-CRC-1 cells showed higher expression of Na_V_1.7 compared to the weakly metastatic TT cells (Figure 3A). An additional MTC specimen analysis revealed that the expression of Na_V_1.7 was found in four of the six patient tissues that were examined, while it was not detected in the normal thyroid specimen (Figure 3B). We also detected the presence of somatostatin receptor, SSTR2, a known MTC biomarker, in four patient tumor tissues, of which three had the presence of Na_V_1.7 expression (Figure 3C) [71].

Overall, the results from quantitative PCR and immunoblotting showed that the expression of Na_V_1.7 was found in all MTC cells and in 66.7% (four out of six) of the patient tissues that were examined, while it was not detected in any normal thyroid specimens. The highly metastatic MZ-CRC-1 cells showed higher expression of Na_V_1.7 compared to the weakly metastatic TT cells. These results are consistent with recent reports of Na_V_1.7 mRNA expression in the prostate cancer cell lines in rat, MAT-LyLu and AT-2, and human, PC-3 and LNCaP. The cell lines with stronger metastatic potential (MAT-LyLu and PC-3) had 1000-fold higher Na_V_1.7 expression than the weakly metastatic cell lines (AT-2 and LNCaP) [66]. Moreover, the study of Na_V_1.7 expression in human prostate biopsies demonstrated that Na_V_1.7 expression was elevated in prostate cancer samples (~20 fold higher) compared to non-cancerous prostate samples [67]. Similarly, the expression of the neonatal splice variant of Na_V_1.5 (nNa_V_1.5) in breast cancer cells was reported to be lower in the weakly metastatic breast cancer cell line, MCF-7, and higher in the highly metastatic triple-negative breast cell line, MDA-MB-231 [16,72,73].

### 3.2. High-Throughput Analysis of Na_V_1.7 Expression in Human MTC

We further confirmed the overexpression of Na_V_1.7 in a larger set of MTC patients. We constructed tissue microarrays (TMAs) consisting of 45 human samples including normal thyroid and MTC tissues and performed an immunohistochemical (IHC) analysis [74]. The IHC results from the TMAs confirmed that Na_V_1.7 was significantly upregulated in MTC compared to normal thyroid tissue. This result is consistent with the results of our immunoblotting and RT-qPCR analysis. Positive and negative IHC controls were prepared by staining Na_V_1.7 antibody on a MZ-CRC-1 cell pellet (highly expressed Na_V_1.7) and the normal thyroid cell line, a Nthy-ori3-1 cell pellet (no detectable expression of Na_V_1.7) (Figure 4A). A total of 45 tissue samples including normal thyroid, MTC primary, and MTC metastases, distributed on four TMA slides, were prepared and stained with Na_V_1.7 antibody (Figure 4B). The positive and negative expression of Na_V_1.7 were verified with a patient’s metastatic status by pathologists at the UAB Department of Pathology. The quantification of TMAs was carried out through the automated processing of the MTC tissue cores with a custom MATLAB code, which enabled segmentation of the tissue and then histogram analysis with Otsu’s thresholding to separate positive versus negative staining (Figure 4C). Overall, 70.7% of MTC patients showed ≥50% Na_V_1.7 expression (29/41), with a median of 60.37% and a mean of 54.56 ± 1.93% (Figure 4D). There was a statistically significant difference in the percentage of Na_V_1.7 expression between the normal thyroid samples (non-cancerous) and the MTC patient samples (cancerous) (Figure 4E, Appendix A).

The quantitative RT-PCR, immunoblotting, and TMA results suggested that the level of Na_V_1.7 expression in the MTC cell lines and patient tissues could be related to a patient’s metastatic status, and, therefore, we performed a point-biserial correlation on 133 human specimens from all TMAs to determine the relationship between the percentage of Na_V_1.7 expression vs. the disease status; using normal thyroid, primary MTC, and metastatic MTC tissues (Appendix A and Appendix A). We found that there was a positive correlation between the percentage of Na_V_1.7 expression and the patient disease status from normal to metastases, which was statistically significant. However, once we investigated the expression level of Na_V_1.7 in the primary MTC and metastatic MTC samples, the results showed no significant difference between these groups (Figure 4F).

Overall, our results show that the expression of Na_V_1.7 is substantially higher in the MTC cells and MTC patient tissues compared to that of the normal thyroid cells and normal thyroid tissues. Therefore, Na_V_1.7 in MTC could be used as a therapeutic target for drug discovery and/or as a biomarker for diagnostic purposes.

### 3.3. Identification of Na_V_1.7 Inhibitors

Several compounds from our known Na_V_1.5 inhibitor library were used for the initial screening, aimed at identifying Na_V_1.7 inhibitors [40]. This screening resulted in the identification of three potential lead compounds, **SV188**, compound **4**, and **WJB-133** (Figure 5A), for Na_V_1.7 inhibition in MTC. This screening was carried out using the highly metastatic MTC cell line MZ-CRC-1, which has the highest basal expression of Na_V_1.7. The cytotoxicity of the three compounds against MZ-CRC-1 was determined first using an MTT assay. The results revealed that MZ-CRC-1 is more sensitive to **SV188** and **WJB-133** compared to compound **4**. The IC_50_ of **SV188** and **WJB-133** is 9.00 ± 1.92 µM and 8.04 ± 0.47 µM, respectively, whereas compound **4′**s IC_50_ was two-fold higher (Figure 5B). In recent reports, the inhibition of Na_V_1.7 in gastric cancer using TTX significantly reduced the expression of NHE1 at the mRNA and protein levels [34]. In addition, the inhibition of Na_V_1.6 and Na_V_1.7 in prostate cancer cells with small molecules, **S0154** and **S0161**, promoted the degradation of Na_V_ proteins in prostate cancer cells and downregulated both the Na_V_1.6 and Na_V_1.7 protein expression levels, with no significant effect on cell apoptosis at the same concentration [35]. To identify Na_V_1.7 inhibitors using a similar approach, we evaluated the three compounds at a 5 µM concentration after 24 h of treatment for their effects on Na_V_1.7 and related genes in the MZ-CRC-1 cell line. Based on a preliminary screening using RT-qPCR against two genes, *SCN9A* (Na_V_1.7) and *SLC9A1* (NHE1), compound **SV188** was selected for further evaluation, as it substantially lowered the expression of *SLC9A1* (NHE1) and *SCN9A* (Na_V_1.7) compared to the corresponding controls (Figure 5C).

Next, we tested **SV188** against the Na_V_1.5, Na_V_1.6, and Na_V_1.7 channels that have been reported to be involved in cancer cells’ migration and invasion [41,70]. The results showed that the treatment of **SV188** at 5 µM for 48 h significantly decreased the mRNA expression of Na_V_1.7 and increased the mRNA expression of Na_V_1.5, with no significant effect on the mRNA expression of Na_V_1.6 (Figure 6). Although the treatment of **SV188** affected Na_V_1.5 expression, when comparing the expression of all three channels, the expression of Na_V_1.7 was 400-fold higher than that of Na_V_1.5 and 25-fold higher than that of Na_V_1.6 (Figure 2A); the effect on Na_V_1.7 is more likely to outweigh the effect on Na_V_1.5.

### 3.4. Synthesis of Compound **4**, **SV188** and **WJB-133**

Compound **4** was synthesized using a previously reported procedure from our lab [40]. The synthesis of the compound **SV188** was carried out in three steps starting from γ-phenyl-γ-butyrolactone (**1**), as outlined in Figure 1. Lactone **1** was first converted to 4,4-diphenylbutyric acid (**2**) by treatment with AlCl_3_ in anhydrous benzene with a 94% yield. Carboxylic acid **2** was converted to amide **6** using the EDC-mediated amide coupling reaction with 3-piperidylpropanamine (**5**) with an 83% yield. The amine **5** used in the amide coupling reaction was obtained with an 89% yield by the reduction of 3-piperidylpropionitrile (**3**) using Raney Ni in MeOH. Reduction of amide **6** with LiAlH_4_ in THF, followed by the conversion of the product amine to its hydrochloride salt by treatment with HCl, afforded **SV188** as a hydrochloride salt with a 61% yield in two steps.

Compound **WJB-133** was synthesized in two steps, as shown in Figure 2. Carboxylic acid **2** was converted to amide **8** using the EDC-mediated amide coupling reaction with 3-phenylpropanamine (**7**) with a 73% yield. Reduction of amide **8** with LiAlH_4_ in THF, followed by the conversion of the product amine to its hydrochloride salt by treatment with HCl, afforded **WJB-133** as a hydrochloride salt with a 47% yield in two steps.

### 3.5. Dose-Dependent Inhibition of Na_V_1.7 Currents (INa) by **SV188**

To test the ability of **SV188** to inhibit the sodium currents (*I*_Na_) carried by the Na_V_1.7 channel, we performed whole-cell patch-clamp experiments using HEK-293 cells transiently expressing Na_V_1.7 and measured the dose-dependent inhibition of the *I*_Na_ peak evoked by depolarizations to −10 mV from a holding potential (HP) of −120 mV applied every 10 s. We first tested the effect of 0.06% DMSO alone on the *I*_Na_ amplitude in all the patch-clamped cells and found that DMSO diminished the current magnitude by 3% on average. Then, the cells were superfused with increasing concentrations of **SV188**, and the *I*_Na_ peak currents were measured (Figure 7A). A stationary blockade of *I*_Na_ was reached around 4–5 min after superfusing the cell with each concentration of the compound (Figure 7B). The inhibition of *I*_Na_ by **SV188** was partially reversed (74%) after washing with control saline for a period of 10–15 min. The fraction of *I*_Na_ unblocked by **SV188** in each cell was averaged and plotted as a function of the compound concentration, and the data were fitted with the Hill equation (Figure 7C). To determine whether the **SV188** blockade of *I*_Na_ was more effective at more depolarized holding potentials, we investigated the effect of **SV188** at 3 µM and 10 µM on the partially inactivated channels by using an HP of −80 mV. The fraction of the sodium current that was blocked at −10 mV under these experimental conditions was practically the same as with an HP of −120 mV (solid pink circles in Figure 7C), and both data points overlap with the fit of the data obtained with an HP of −120 mV. These results suggest that **SV188** inhibits the *I*_Na_ with the same potency in the closed state (HP of −120 mV) and the inactivation state (HP of −80 mV) of the Na_V_1.7 channels [75,76].

### 3.6. Effects of **SV188** on Na_V_1.7 Channels Gating

The current–voltage relationships (*I*–*V* curves) for Na_V_1.7 channels were measured using 16-ms step depolarizations to varying potentials from an HP of −60 to +100 mV in 10 mV steps. The representative families of the *I*_Na_ recordings obtained from an HEK-293 cell expressing Na_V_1.7 channels in the absence of, during, and after exposure to 5 µM of **SV188** are shown in Figure 8A. The average *I–V* curves are shown in Figure 8B. The maximum peak current was observed at −10 mV under control recording conditions, and the blockade by **SV188** shifted this value to −20 mV. This effect was also observed in the recordings shown in Figure 8A. To further analyze the effect of **SV188** on the voltage-dependent activation of the Na_V_1.7 channels, we calculated the channel conductance with the equation G(*V*) = *I*/(*V* − *V*_rev_), where *I*, *V*, and *V*_rev_ represent the sodium current elicited (as shown in Figure 8A), test potential, and reversal potential, respectively. The conductance values were normalized and plotted as a function of the test potential for the Na_V_1.7 channels in the absence and presence of 5 µM of **SV188**, and each data set was fitted to a Boltzmann function (Figure 8C, smooth lines). The obtained parameters indicate that **SV188** shifted the voltage-dependence of the Na_V_1.7 channels’ activation to more negative potentials by 8.5 mV. In addition, although **SV188** effectively blocked *I*_Na_ over a wide range of testing potentials, it was clearly more potent at more positive voltages, being more evident for the *V*_m_ values beyond the *V*_rev_ (Figsure 8A,B). For example, at −10 mV, 5 µM **SV188** inhibited *I*_Na_ by an average of 56%. By comparison, at +80 mV, *I*_Na_ was inhibited by 92% (Appendix A). These results suggest that the inhibition of the Na_V_1.7 sodium current by **SV188** is voltage-dependent, with a stronger block at membrane potentials where the *I*_Na_ should be outward.

We next sought to determine whether **SV188** alters the voltage-dependence of the Na_V_1.7 channels’ inactivation. For this purpose, we used a classical two-pulse voltage clamp protocol. The first step was a 200 milliseconds prepulse to voltages between −120 and −50 mV, intended to promote channels into an inactivated state. The second voltage step was a brief test pulse to −10 mV, in which the relative amplitude is proportional to the fraction of Na_V_ channels that were not inactivated by the prepulse. The representative *I*_Na_ obtained from the Na_V_1.7 channel recorded in the absence and presence of 5 µM **SV188** is illustrated in Figure 8D. It is shown that in the absence and presence of **SV188**, the current amplitude at −10 mV after a prepulse to −90 mV is roughly the same, i.e., around 63% of the maximal current in each condition. This observation was further analyzed with the inactivation curves shown in Figure 8E. The normalized data of the *I*_Na_ recorded during the test pulses to −10 mV was plotted as a function of the prepulse potential. The data points were well-fitted by single Boltzmann functions, assuming that the channels fully inactivated at depolarized voltages. In this case, the interaction of **SV188** with the Na_V_1.7 channels led to a non-significant shift of 7 mV in the voltage dependence of inactivation toward more negative potentials, suggesting that the inactivated state of the channel is not affected by **SV188** binding, and vice versa. This result is also consistent with the observation that the percentage of *I*_Na_ that was blocked is not different when using an HP of −120 or −80 mV (Figure 7C). Interestingly, in a previous work by our group, several secondary amine compounds that have a similar chemical structure to **SV188** induced a significant state-dependent effect on the Na_V_1.5 sodium currents of the MDA-MB-231 breast cancer cell line [40]; however, the potential use-dependence effect was not explored for such compounds. It is likely that the lack of a state-dependent effect of the **SV188** in the Na_V_1.7 channels could be due to a discrete difference in the sequence/structure when compared with the Na_V_1.5 channels.

### 3.7. Use-Dependent Blockade of Na_V_1.7 Channels by **SV188**

VGSC inhibitors such as local anesthetics, antiarrhythmics, and opiate antihyperalgesics are known to display state-dependent and use-dependent channel blockades [47,75,77,78,79]. This characteristic constitutes a functional selectivity for inhibitors to preferentially bind to channels that are activated frequently, thus attracting additional molecules to bind to such states. To further investigate the inhibition of the resting state of Na_V_1.7 channels by **SV188**, the currents were first recorded at 10-s intervals in a control solution before superfusing the cell with 5 µM of **SV188** for 5 min without applying any depolarizing steps (Figure 9A). When the voltage steps were resumed, the *I*_Na_ was initially inhibited by 20% (Figure 9A–C, p1). However, the proportion of inhibition increased with subsequent test pulses, reaching a maximum of 77% at 4.5 min (Figure 9A–C, p*n*). Thus, the inhibition of Na_V_1.7 channel currents by **SV188** requires the opening of the channel for the binding of the compound to its site of action. Although it was shown that fenestrations of VGSCs could be an alternative gateway to the central cavity of some resting-state blockers [80], this could not be the preferred shortcut for **SV188** on the Na_V_1.7 channel, as only a very small fraction of channels were blocked at −120 mV, suggesting that the main access route for **SV188** to its binding site should be the intracellular gate of the Na_V_1.7 channel [81]. On the contrary, tetrodotoxin (**TTX**), a well-known open channel blocker of VGSCs [42,82,83], did not need the channel to be opened to induce the maximum blockade of the *I*_Na_, as the first depolarizing pulse after resuming the voltage steps practically showed the same blocked fraction of the *I*_Na_ as that reached at the stationary blockade (Figure 9B,C, p1, p*n*). The observation that the compound needed periodic depolarization to block the *I*_Na_ suggests that the channels’ states visited during the depolarizations unmasks higher affinity conformations compared with the closed state.

To further test the use of the dependence effect of **SV188** on Na_V_1.7 channels, cells were held at −120 mV, and sodium currents were elicited by a train of 40 16-ms pulses to −10 mV at 40 Hz. The peak current amplitude at each pulse was normalized to that of the first pulse. As shown in Figure 9D, **SV188** displayed preferential inhibition on test pulse 40 compared to pulse 1, showing an 81% current decrease (blue points); whereas, in the absence of the blocker, the level of the current decrease is only 30% from pulse 1 to pulse 40 (black points), which implies that the blockade of Na_V_1.7 channels by **SV188** is increased by around 50% when the channel is activated at 40 Hz, in comparison to when the channels are activated every 10 s (0.1 Hz; Episode 1, blue points, Figure 9D).

The results of the electrophysiological studies presented here suggest that **SV188** is a dose-dependent and voltage-dependent inhibitor. The *I*_Na_ blockage was greater at higher concentrations of **SV188**, and the inhibition of the Na_V_1.7 channel is stronger at more depolarized membrane potentials. These studies also suggest that **SV188** functions as a use-dependent inhibitor of Na_V_1.7 because the highest percentage of inhibition of *I*_Na_ by **SV188** was observed when the channels were activated at higher frequencies (Figure 9D).

### 3.8. Effect on MTC Cell Viability by **SV188**

The results of the electrophysiological study suggested that **SV188** is a use-dependent blocker of Na_V_1.7 at low micromolar concentrations and that the observed effects are reversible, highlighting the potential for **SV188** to inhibit the migration and invasion activities of MTC cells. A highly aggressive MTC cell line originated from lymph node metastasis, MZ-CRC-1, and a less-aggressive MTC cell line, TT, derived from the primary tumor, were used for the cell migration and invasion inhibition studies. These two cell lines are the only available human MTC-derived cells. These studies needed to be conducted at lower doses than the cytotoxic concentrations of **SV188** to ensure that the observed effects on cell migration and invasion were independent of the effects on the cell viability. Therefore, the inhibitory effects of **SV188** on the MTC cell lines (IC_50_ values), MZ-CRC-1 and TT, were determined using the reported MTT assay [84]. The MTC cells were treated in quadruplicate for each concentration in each individual experiment. The results from each experiment were plotted as a normalized curve fit vs. a dose response (variable slope) to obtain the IC_50_ value. The experiments were repeated three times, and the average IC_50_ value was calculated as mean ± SEM. **SV188** inhibited the cell viability of the MZ-CRC-1 cells, with an IC_50_ value of 8.47 μM, and the TT cells, with an IC_50_ value of 9.32 μM (Figure 10A).

### 3.9. Effect on Cell Migration by **SV188**

The migration- and invasion-inhibitory activities of **SV188** were evaluated using MZ-CRC-1 and TT cells in a reported Boyden Chamber assay [85] at two doses (3 µM and 6 µM) lower than its cell viability IC_50_ value. In the Boyden Chamber assay, the ability of cancer cells to invade is measured based on the number of cells that can invade the matrigel and migrate through the pores across the membrane. The MZ-CRC-1 and TT cells were treated with 3 μM and 6 μM of **SV188** and compared to control 0.06% DMSO for 48 h, and the number of invade cells was counted manually. Our results revealed that at 3 μM of **SV188**, the dose significantly reduced the MTC cells’ migration by 27% and 57% for MZ-CRC-1 and TT cells, respectively. The percentage of inhibition on the cell migration in MZ-CRC-1 increased to 42% when treated with 6 μM of **SV188**. However, the degree of migration inhibition of TT at 6 μM was relatively similar as that at 3 μM: 53% vs. 57%, respectively (Figure 10B,C).

### 3.10. Effect on Cell Invasion by **SV188**

**SV188** significantly inhibited MZ-CRC-1 cell invasion by 35% and 52% after treatment with 3 μM and 6 μM, respectively. In contrast, **SV188** showed no effect on the invasion of TT cells with lower basal expression of Na_V_1.7 when derived from the primary tumor (Figure 10D,E). The lack of invasion inhibition by **SV188** in TT cells may result from the weakly metastatic potential and low expression of Na_V_1.7. Similar results were found in a previous report of the comparison invasion inhibition of the weakly metastatic breast cancer cell line MCF-7 and the highly metastatic breast cancer cell line MDA-MB-231, where the MCF-7 cell line showed no response to Nav1.5 inhibitor (phenytoin) treatment in contrast to MDA-MB-231, where treatment substantially reduced cancer cell invasion [73]. MZ-CRC-1 cells, which have significantly higher basal expression of both Na_V_1.5 and Na_V_1.7 and originated from a lymph node metastasis, showed a reduction in both migration and invasion. In contrast, **SV188** treatment of TT cells, which were derived from the primary tumor and expressed lower basal levels of Na_V_1.5 and Na_V_1.7, only inhibited migration. Our results suggest that the MTC cell invasion inhibition by **SV188** is directly correlated with the expression level of the sodium channels in these cells.

### 3.11. Cell Cycle Analysis in Response to **SV188** Treatment

We performed a flow cytometry analysis to investigate the effects of **SV188** on the MZ-CRC-1 cell cycle. This study revealed that **SV188** induced cell cycle arrest at the G0/G1 phase and decreased the cell population at the S and G2 phases (Figure 11). Voltage-gated ion channels (VGICs) play an important role in cell cycle progression through the differentiation of the membrane potential (*V*_m_). Cells in the resting state have more negative *V_m_* compared to cells during proliferation. Additionally, *V*_m_ becomes less negative or depolarized due to the transition from the G0/G1 phase to the S phase, and VGSCs and/or Ca^2+^ channels are opened, resulting in a positive (+) ions influx inside the cells. Then, VGSCs and/or Ca^2+^ channels are close during the S-phase, causing *V*_m_ repolarization leading back to an initial phase of the cell cycle, G0/G1 [86,87,88,89]. Therefore, the inhibition of VGSCs could potentially affect cell cycle arrest and inhibit cell proliferation. There are a few studies that explored the effect of VGSC inhibitors on the cell cycle, such as a report from Li et al. in 2018 indicating that three out of six VGSC drugs, levobupivacaine (25 µM), ropivacaine (35 µM), and chloroprocaine (150 µM), inhibited cell migration in a wound healing assay after 24 h in human breast cancer MDA-MB-231 cells that expressed Na_V_1.5 [90]. From this cell cycle analysis, levobupivacaine and chloroprocaine slightly activated cell cycle arrest at the S phase, while ropivacaine remarkably induced cell cycle arrest at the G2/M phase [90]. Interestingly, lidocaine, a VGSC inhibitor that was reported to decrease cell proliferation and reduce cancer cells’ migration and invasion [64,91,92], showed a mild effect on cell cycle arrest at the S phase, with no significant influence on the migration of MDA-MB-231 cells at its antiarrhythmic plasma concentration (10 µM) after 24 h of treatment [90]. Additionally, the treatment of lidocaine at 100 µM was reported to inhibit cell growth at 72 h and increase apoptosis at 48 h; however, it did not show a significant effect on cell cycle arrest in hepatocellular carcinoma HuH7 and HepaRG cells (both cell lines had no report of VGSCs’ expression) [93]. A recent study on the treatment of lidocaine in cervical cancer HeLa cells that expressed Na_V_1.6 [65] found that this drug significantly inhibited the cell growth at 0.3 mM by reducing a proliferating protein, Ki-67 (MKI67), and a cell cycle analysis indicated that lidocaine significantly induced arrest at the G0/G1 phase and decreased cells’ population at the G/M and S phases in a dose-dependent manner [91]. In addition, the knockdown of Na_V_1.5 in oral squamous cell carcinoma (OSCC) HSC-3 cells caused cell cycle arrest at the G1 phase and a drastic reduction in cell migration and invasion [94]. The cell progression in knockdown Na_V_1.5 HSC-3 was reported to be regulated by the Wnt/β-catenin signaling pathway, which also had an influence on cancer cell migration and invasiveness [95,96,97].

In this current study, we saw the effect of **SV188** on cell cycle arrest at the G0/G1 phase, which could lead to the inhibition of cell proliferation by inducing cell apoptosis, as observed in previous reports on the induction of apoptosis associated with Nav1.5 and Nav1.6 expression with siRNAs in astrocytoma [98], the expression of neonatal Nav1.5 in human brain astrocytoma, and its effect on the proliferation, invasion, and apoptosis of astrocytoma cells [99] and follicular thyroid carcinoma cells [53]. We also noticed a significant decrease in the mRNA expression of Na_V_1.7 and NHE-1 and a reduction in cell migration and invasion after treatments with **SV188**. The mechanism of how inhibition of sodium channels inhibits cancer metastases have not been fully elucidated. However, a plausible mechanism pathway for this effect could involve VGSCs’ colocalized proteins such as NCX and NHE-1, as shown in the schematic in Figure 1C [33,36,70]. One of the important factors contributing to the metastasis is the ability of highly aggressive cancer cells to cause proteolytic degradation of the extracellular matrix (ECM), break away from the tumor site, enter the bloodstream, and travel to distant sites to initiate metastasis. Recent literature showed that nNav1.5 activity in MDA-MB-231 cells enhances ECM degradation [100] by activating cysteine cathepsins B and S through the acidification of the pericellular microenvironment [38,101]. The Na^+^/H^+^ exchanger (NHE1) is the central regulator of intracellular and perimembrane pH, which is also overexpressed and overactivated in cancer cells [102,103]. This acidity activates cathepsins and proteolytic degradation of the ECM [104]. Thus, the persistent activity of nNav1.5 at the membrane potential of breast cancer cells (about −36 mV) is responsible for increased ECM proteolysis and cancer cell invasion [38,105]. Moreover, the changes in sodium level across the cell membrane produced by Nav1.7 inhibition may activate the function of NCX and NHE-1 proteins. Several studies disclosed that the reduction in cell migration and invasion was caused by the decrease in calcium-dependent proteins that are essential for epithelial–mesenchymal transition (EMT) and the reduction of H^+^ efflux through NHE-1 [55,106,107,108]. Therefore, in addition to the downstream effect on NHE-1, in future studies it would also be interesting to investigate the changes in calcium-dependent proteins (N-cadherin, vimentin, and snai1) and the changes in cysteine cathepsins’ activity between **SV188**-treated and -untreated MTC cells.

## 4. Conclusions

In conclusion, we reported, for the first time, the overexpression of Na_V_1.7 (*SCN9A* gene) in aggressive and metastatic MTC as a potential target for drug discovery. Our results from quantitative RT-PCR, Western blotting, and the TMA immunostaining of 45 patient specimens, including both normal thyroid and MTC samples, confirmed that the VGSC subtype Na_V_1.7 was specifically overexpressed in MTC, while it was not expressed in normal thyroid cells and tissues. A highly metastatic cell line, MZ-CRC-1, originating from a lymph node metastasis, showed a remarkably high expression of Na_V_1.7 compared to the low-level expression in TT cells derived from the primary tumor, suggesting a role for Na_V_1.7 in MTC metastasis. We demonstrated the druggability of Na_V_1.7 in MTC, by identifying a novel inhibitor (**SV188**) of this channel, and investigate its mode of binding and its ability to block the Na_V_1.7 sodium current. Patch-clamp studies of **SV188** in the Na_V_1.7 channels expressed in **HEK**-293 cells showed that **SV188** inhibited the Na_V_1.7 current with an IC_50_ value of 3.6 µM and Hill coefficient of 1.2. The results of our electrophysiological studies suggested that **SV188** blocks the Na_V_1.7 channel in a voltage- and use-dependent manner, without significant effects on the steady-state inactivation of the channel. The inhibition of *I*_Na_ by **SV188** led to a significant shift in the Na_V_1.7 channel’s conductance activation to more hyperpolarized potentials (around 8 mV). The mechanism of the blocking of Na_V_1.7 channels by **SV188** did not involve an effect on the steady-state inactivation nor did the percentage of *I*_Na_ that is blocked show any differences when using different HPs. In addition, our results demonstrated that a higher blockade of outward *I*_Na_ agrees with the use-dependence effect of **SV188** in the Na_V_1.7 channels, as Na^+^ ions moving out of the cell found the pore channel pathway blocked by the presence of **SV188**, which is favored by the higher frequencies of the channel openings. Altogether, the electrophysiological data suggested that **SV188** might be entering the central cavity of the channel through the intracellular gate and binding somewhere in the permeation pathway of the channel. **SV188** inhibited the viability of two MTC cell lines, MZ-CRC-1 and TT, with IC_50_ values of 8.47 μM and 9.32 μM, respectively. Supporting our hypothesis, **SV188** significantly inhibited the invasion of MZ-CRC-1 cells by 35% and 52% after treatment with 3 μM and 6 μM, respectively. In contrast, **SV188** showed no effect on the invasion of TT cells derived from the primary tumor, which has lower basal expression of Na_V_1.7. **SV188** significantly inhibited the cell migration of MZ-CRC-1 and TT cells by 27% and 57%, respectively, at a 3 μM concentration. The dose at which **SV188** displayed an inhibition of the invasion and migration of MTC cells was below their cell viability IC_50_ values, indicating that these effects are independent from the drug cytotoxicity. In addition, the cell cycle analysis of MZ-CRC-1 indicated that the treatment of **SV188** induced arrest at the G0/G1 phase and decreased the cell population at the S and G2 phases, which led to the inhibition of MZ-CRC-1 cell proliferation, possibly by promoting cell apoptosis. Overall, our data showed that Na_V_1.7 is uniquely overexpressed in MTC and suggested that Na_V_1.7 could serve as a target to develop small-molecule drugs and/or as a biomarker for diagnostic purposes. It is reported that individuals carrying a mutation in SCN9A do not express Na_V_1.7 in their cells and have a congenital insensitivity to pain (CIP), a rare autosomal recessive disorder in which affected individuals are unable to perceive pain from birth to death [109,110]. Therefore, an additional benefit of using Na_V_1.7 inhibitors in cancer therapy would be their ability to reduce cancer-related pain. Studies to examine **SV188** and other Na_V_1.7 inhibitors as potential pain therapeutics are currently in progress [76,111].

## Data Availability

Data is contained within the article or Appendix A.

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
