# Peer review of "Voltage-Gated Sodium Channel NaV1.7 Inhibitors with Potent Anticancer Activities in Medullary Thyroid Cancer Cells"

_cancers, 2023, doi:10.3390/cancers15102806_

Round 1
Reviewer 1 Report
Pukkanasut et al. report on a novel NaV1.7 VGSC blocker, SV188, and its efficacy in slowing the growth, motility, and invasiveness of medullary thyroid cancer (MTC), which over-expresses this subtype of VGSCs. Because SV188 slow these dependent measures that have been correlated to cancer aggression and metastasis, it may be an efficacious treatment for MTC. The manuscript is well written with a good introduction to VGSCs as a target for cancer treatment. The statistics are appropriate and are not over-interpreted. The electrophysiological studies are thoughtful and thorough. The only data that this reviewer would like to see is a survival study in a homograft or xenograft model. However, this may be more appropriate for a second manuscript. A few minor critiques: 1) in the introduction the authors refer to cancer cells as non-excitable, but cancer cells that over-express VGSCs have been shown to be excitable; they will spontaneously discharge. 2) On lines 954-956, the authors conclude that SV188 enters the channel from through the intracellular gate. Is SV188 a non-polar molecule so that it can cross the cell membrane to access the channel, or is there an active transport mechanism? 3) The Conclusions section would be more complete if the authors discussed that by preventing progression from G0/G1 to S phase of the cell cycle this would send the cells into the apoptosis pathway. 4) Humans that do not express NaV1.7 channels are anhedonic. You may want to consider that blockade of NaV1.7 channels may feel no pain at all.
Grammatical:
Line 256: Please rewrite this sentence. It reads as if the cells are allowing themselves to grow.
Line 517: the two semi-colons should be colons.
Line 530: All other instances of “Nthyori” were hyphenated
Author Response
We would like to thank the reviewers for their careful review and thoughtful critique of our manuscript. Our responses to their comments are as follows.
Reviewer 1
R1a. In the introduction the authors refer to cancer cells as non-excitable, but cancer cells that over-express VGSCs have been shown to be excitable; they will spontaneously discharge.
Response: The excitable cells are the cells that have the ability to be electrically excited resulting in generation of the action potentials [1]. We believe that there is not enough evidence in the literature for cancer cells to be defined as excitable cells. Although there are studies that show a large membrane currents and Vm fluctuations in highly metastatic triple negative breast cancer cells (MDA-MB-231) [2,3], such discrete fluctuations might not be enough to generate or propagate action potentials. The biophysical parameters such as small current density and depolarized Vm, where most of the VGSCs are already inactivated, do not trigger the generation of canonical electrical signals in cancer cells. Nevertheless, it could be expected that the overexpression of VGSCs in cancer cells would induce a change in the electrical properties of these cells compared to non-cancerous cells. We have explained the excitability of the cancer cells in the introduction section of the revised manuscript (lines: 80-84).
R1b. On lines 954-956, the authors conclude that SV188 enters the channel from through the intracellular gate. Is SV188 a non-polar molecule so that it can cross the cell membrane to access the channel, or is there an active transport mechanism?
Response: Generally, local anesthetic drugs containing secondary and tertiary amine groups similar to our inhibitors diffuse through the cell membrane uncharged and get protonated under physiological conditions before reaching its binding site [4]. These drugs are active in their protonated forms [4,5]. Compound SV188 in its protonated form has a predicted pKa values of 7.9 and 10.5 (ChemAxon) [6] (Fig. 1). At physiological condition, SV188 is expected to be protonated. However, SV188 also has a hydrophobic part in the structure, which makes it amphiphilic. The predicted LogP value of SV188 by SeeSAR is 3.65 [7] suggesting that this compound is typically lipophilic and can permeate the cell membranes [8,9]. Sodium channel inhibitors have two main ways to access its binding site, through the extracellular gate or through the intracellular gate. Most protonated amine drugs enter the channel when the gate is open through the intracellular gate whereas the basic amine with uncharged can directly diffuses through the plasma membrane (passive transport) [5]. Therefore, the predominant lipophilic nature of SV188 agrees with our electrophysiological data, which suggests its main entry to the channel through the intracellular gate.
Figure 1. Structure of SV188.
R1c. The Conclusions section would be more complete if the authors discussed that by preventing progression from G0/G1 to S phase of the cell cycle this would send the cells into the apoptosis pathway.
Response: We have added a discussion about the potential effect of the prevention of the cell cycle progression from G0/G1 to S phase leading to cell apoptosis in the cell cycle results section of the revised manuscript (lines: 945-948 and lines: 1023-1026).
R1d. Humans that do not express NaV1.7 channels are anhedonic. You may want to consider that blockade of NaV1.7 channels may feel no pain at all.
Response: An additional benefit of using Nav1.7 inhibitors in cancer therapy is their ability to reduce pain. However, the effect on pain is not the primary focus of this study. A brief discussion on this potential benefit is included in the conclusion (line: 1029-1034).
R1e. Line 256: Please rewrite this sentence. It reads as if the cells are allowing themselves to grow.
Response: This change has been made in the revised manuscript (line: 272).
R1f. Line 517: the two semi-colons should be colons.
Response: This change has been made in the revised manuscript (line: 538).
R1g. Line 530: All other instances of “Nthyori” were hyphenated.
Response: This change has been made in the revised manuscript (line: 189, 492, 551, 565, and 580).
References
- Miura, R.M. Analysis of excitable cell models. J. Comput. Appl. Math. 2002, 144, 29-47, doi:10.1016/S0377-0427(01)00550-7.
- Quicke, P.; Sun, Y.; Arias-Garcia, M.; Beykou, M.; Acker, C.D.; Djamgoz, M.B.A.; Bakal, C.; Foust, A.J. Voltage imaging reveals the dynamic electrical signatures of human breast cancer cells. Commun Biol 2022, 5, 1178, doi:10.1038/s42003-022-04077-2.
- Ribeiro, M.; Elghajiji, A.; Fraser, S.P.; Burke, Z.D.; Tosh, D.; Djamgoz, M.B.A.; Rocha, P.R.F. Human Breast Cancer Cells Demonstrate Electrical Excitability. Front Neurosci 2020, 14, 404, doi:10.3389/fnins.2020.00404.
- Gamal El-Din, T.M.; Lenaeus, M.J.; Zheng, N.; Catterall, W.A. Fenestrations control resting-state block of a voltage-gated sodium channel. Proc Natl Acad Sci USA 2018, 115, 13111-13116, doi:10.1073/pnas.1814928115.
- Hille, B. Local anesthetics: hydrophilic and hydrophobic pathways for the drug-receptor reaction. J Gen Physiol 1977, 69, 497-515, doi:10.1085/jgp.69.4.497.
- ChemAxon. Marvin version 21.9.0. 2021.
- BioSolveIT GmbH, S.A., Germany. SeeSAR version 12.1.0. BioSolveIT 2022.
- Soliman, K.; Grimm, F.; Wurm, C.A.; Egner, A. Predicting the membrane permeability of organic fluorescent probes by the deep neural network based lipophilicity descriptor DeepFl-LogP. Sci Rep 2021, 11, 6991, doi:10.1038/s41598-021-86460-3.
- Oruç, T.; Küçük, S.E.; Sezer, D. Lipid bilayer permeation of aliphatic amine and carboxylic acid drugs: rates of insertion, translocation and dissociation from MD simulations. Phys. Chem. Chem. Phys. 2016, 18, 24511-24525, doi:10.1039/C6CP05278A.

Reviewer 2 Report
Voltage-gated sodium Channel Nav1.7 Inhibitors with Potent anticancer activities in Medullary Thyroid cancer cells
In this paper, the authors examine the mRNA and protein levels of Channel Nav1.7 and found that this protein is overexpressed in MTC cells and patients’ samples. They further show that SV188 behaves as an inhibitor of migration and invasion of aggressive treated MTC cells.
The paper is nicely written and clear.
Major concerns:
1- The specificity of SV188 should be addressed by repeating the same experiments as in Figure 5C with unrelated channels, figure 7, in other cell migration and invasion assays….
2- By which molecular/ cellular mechanism the inhibitor would alter/reduce migration or invasion of cancer cells? Because an important number of studies have reported the involvement of Nav channels in human cancers so how would they promote MTC migration and or invasion by which pathways? This is not discussed while it appears to be essential to the readers. The observed decrease in migration would it be relevant physiologically? Authors should compare and discuss these points and provide a working model.
Minor points
Figure 1 could be used with the working model at the end.
Figure 2: On panels, B and C it should be written relative fold gene expression of Nav1.7 since in A both are written Nav1.5 and Nav1.7
Lane 506 a space is missing: showed higher
Figure 5 lane 620 Nav1.2 (SCN2A) levels are not shown in panel C
Author Response
Reviewer 2
We would like to thank the reviewers for their careful review and thoughtful critique of our manuscript. Our responses to their comments are as follows.
R2a. The specificity of SV188 should be addressed by repeating the same experiments as in Figure 5C with unrelated channels, figure 7, in other cell migration and invasion assays.
Response: We do not expect our compounds including SV188 to be highly specific inhibitors of individual channels. As presented in the manuscript, we identified our lead Nav1.7 inhibiting compound, SV188 by screening of a NaV1.5 inhibitor library [1] against MTC cell lines. We describe SV188 as a NaV1.7 inhibitor due to its ability to block NaV1.7 channel that is highly expressed in MTC cell lines (> 400-fold higher compared to NaV1.5, Fig 2A).
Based on our proposed mechanism (Fig 1C), the observed reduction in migration / invasion potentially is a result of NaV1.7 inhibition by SV188. We performed quantitative RT-PCR experiments using MZ-CRC-1 cells to detect the effect of SV188 treatment on other Nav1 channels that have been reported to be involved in cancer cell migration and invasion: NaV1.5, NaV1.6 and NaV1.7 [2-4]. The results showed that the treatment of SV188 significantly decreased mRNA expression of NaV1.7 and increased mRNA expression of NaV1.5 with no significant effect on mRNA expression of NaV1.6. Although the treatment of SV188 affected NaV1.5 expression compared to the expression of other two channels, the expression of NaV1.7 was 400-fold higher than NaV1.5 and 25-fold higher than NaV1.6. The effect on NaV1.7 is more likely to outweigh the effect on NaV1.5. To emphasize the differences on the expression level of Nav1.5, Nav1.6 and Nav1.7 in NET cell lines, we performed RT-qPCR experiment on MZ-CRC-1, TT, BON and H727 cells against Nav1.5, Nav1.6 and Nav1.7 and revised the manuscript under the VGSC expression in neuroendocrine tumors (NETs) (line: 475-486). We also revised the identification of NaV1.7 inhibitor in the result section to show the effect of SV188 on NaV1.5, NaV1.6 and NaV1.7 expression (line: 647-655).
We thank reviewer 2 for his/her suggestion about conducting the cell migration and invasion assays in the other cell lines that do not express Nav1.7. However, there are no MTC cell lines currently available that do not express Nav1.7. MZ-CRC-1 and TT cells are the only two available MTC cell lines, and we are using both these cell lines in this study. Highly metastatic MZ-CRC-1 cells with a 2-fold higher expression of NaV1.7 compared to the weakly-metastatic TT cells. We have shown that SV188 significantly inhibited invasion of MZ-CRC-1 cells which have high level of Nav1.7 (p<0.01) and originate from lymph node metastasis. Opposite, SV188 did not have effect on invasion of TT cells, which were derived from primary tumor and had lower Nav1.7 level. (Fig. 10 B-E). We conclude that expression of Nav1.7 is associated in vitro with strong invasive behavior in the highly metastatic MZ-CRC-1 cells, whereas the weakly metastatic line, TT cells, do not invade matrigel and express lower level of Nav1.7. Furthermore, SV188 suppresses Na+ current in Nav1.7- strongly expressing and metastatic MZ cells, thus inhibiting Nav1.7-dependent invasion. TT cells with lower Nav1.7 basal level and derived from primary tumor respond to SV188 treatment by decreased migration however they do not change the invasion potential.
Considering the unavailability of MTC cell lines with no Nav1.7 expression, we believe that the relationship between Nav1.7 blockade by SV188 and inhibition of cell migration /invasion properties can be better established by conducting cell migration and invasion assays with NaV1.7 knockdown (shNaV1.7) MZ-CRC-1 cells, shNaV1.7 MZ-CRC-1 cells with SV188, MZ-CRC-1 with SV188 and compare with MZ-CRC-1 cells with basal Nav1.7 expression levels. We have tried to perform transient transfection of NaV1.7 in MZ-CRC-1. Unfortunately, the expression started to come back after 24 h (Fig 2C) and we need 48 h for the migration/invasion assays. We are currently growing MZ-CRC-1 cells transfected with NaV1.7 shRNA. To obtain the stable transfected and Nav1.7 knockdown model and have enough cells to perform the assay, the process would take more than a month due to the low growing rate of MZ-CRC-1 cells. The doubling time of MZ-CRC-1 is around 12-16 days (CelloPub = CLPUB00385). The stable transfected model will be used in the future study regarding the mechanism of NaV1.7 in MTC. These time-consuming studies are currently in progress, the results of which will be reported in subsequent manuscripts.
Figure 2. Nav1.7 silencing in MZ cells. A). Western blot demonstrating Nav1.7 gene-specific inhibition using two specific shRNA sequences CU6-b (location 4162) and CU6-c (location 2449) compering to non-targeting pool of shRNA in the backbone plasmid as control. B). Densitometry analysis of Nav1.7 protein silencing normalized to GAPDH. C). Time dependent recovery of Nav1.7 after transient transfection. 24h exposure to shRNA resulted with the strongest inhibition. Nav1.7 expression started to recover at 48h post shRNA transfection.
R2b. By which molecular/ cellular mechanism the inhibitor would alter/reduce migration or invasion of cancer cells? Because an important number of studies have reported the involvement of Nav channels in human cancers so how would they promote MTC migration and or invasion by which pathways? This is not discussed while it appears to be essential to the readers. The observed decrease in migration would it be relevant physiologically? Authors should compare and discuss these points and provide a working model. Figure 1 could be used with the working model at the end.
Response: We have revised the discussion section of the manuscript (lines: 950-970) by proposing a working model of the possible mechanism for the reduction in cell migration/invasion in metastatic MTC through the inhibition of NaV1.7 channel. We believe the low micromolar concentrations at which SV188 inhibits cell invasion and migration is physiologically relevant based on the literature reports of less active compounds reducing the breast cancer metastasis in in vivo models [5,6]. Besides, our lead compound has better Nav1.7 inhibition IC50 value than clinically used Nav blockers used as local anesthetics [7] and to treat pain [8].
R2c. Figure 2: On panels, B and C it should be written relative fold gene expression of Nav1.7 since in A both are written Nav1.5 and Nav1.7
Response: This change has been made in the revised manuscript (Fig. 2B-C).
R2d. Lane 506 a space is missing: showed higher.
Response: This change has been made in the revised manuscript (line: 526-527).
R2e. Figure 5 lane 620 Nav1.2 (SCN2A) levels are not shown in panel C.
Response: It was a typo, there was no NaV1.2. We apologize for this mistake.
References
- Dutta, S.; Lopez Charcas, O.; Tanner, S.; Gradek, F.; Driffort, V.; Roger, S.; Selander, K.; Velu, S.E.; Brouillette, W. Discovery and evaluation of nNa(v)1.5 sodium channel blockers with potent cell invasion inhibitory activity in breast cancer cells. Bioorg Med Chem 2018, 26, 2428-2436, doi:10.1016/j.bmc.2018.04.003.
- Angus, M.; Ruben, P. Voltage gated sodium channels in cancer and their potential mechanisms of action. Channels (Austin) 2019, 13, 400-409, doi:10.1080/19336950.2019.1666455.
- Horne, J.; Mansur, S.; Bao, Y. Sodium ion channels as potential therapeutic targets for cancer metastasis. Drug Discov Today 2021, 26, 1136-1147, doi:10.1016/j.drudis.2021.01.026.
- Lopez-Charcas, O.; Pukkanasut, P.; Velu, S.E.; Brackenbury, W.J.; Hales, T.G.; Besson, P.; Gomora, J.C.; Roger, S. Pharmacological and nutritional targeting of voltage-gated sodium channels in the treatment of cancers. iScience 2021, 24, 102270, doi:10.1016/j.isci.2021.102270.
- Driffort, V.; Gillet, L.; Bon, E.; Marionneau-Lambot, S.; Oullier, T.; Joulin, V.; Collin, C.; Pagès, J.C.; Jourdan, M.L.; Chevalier, S.; et al. Ranolazine inhibits NaV1.5-mediated breast cancer cell invasiveness and lung colonization. Mol Cancer 2014, 13, 264, doi:10.1186/1476-4598-13-264.
- Nelson, M.; Yang, M.; Dowle, A.A.; Thomas, J.R.; Brackenbury, W.J. The sodium channel-blocking antiepileptic drug phenytoin inhibits breast tumour growth and metastasis. Mol Cancer 2015, 14, 13, doi:10.1186/s12943-014-0277-x.
- Stoetzer, C.; Kistner, K.; Stüber, T.; Wirths, M.; Schulze, V.; Doll, T.; Foadi, N.; Wegner, F.; Ahrens, J.; Leffler, A. Methadone is a local anaesthetic-like inhibitor of neuronal Na+ channels and blocks excitability of mouse peripheral nerves. Br J Anaesth 2015, 114, 110-120, doi:10.1093/bja/aeu206.
- Bok, C.S.; Kim, R.E.; Cho, Y.Y.; Choi, J.S. Tramadol as a Voltage-Gated Sodium Channel Blocker of Peripheral Sodium Channels Na(v)1.7 and Na(v)1.5. Biomol Ther (Seoul) 2023, 31, 168-175, doi:10.4062/biomolther.2023.002.

Round 2
Reviewer 2 Report
The authors addressed my concerns and it is fine